# Early evidence of molariform hypsodonty in a Triassic stem-mammal

Tomaz P. Melo [1], Ana Maria Ribeiro[2], Agustín G. Martinelli [3] & Marina Bento Soares[4]

Hypsodonty, the occurrence of high-crowned teeth, is widespread among mammals with diets rich in abrasive material, such as plants or soil, because it increases the durability of dentitions against wear. Hypsodont postcanine teeth evolved independently in multiple mammalian lineages and in the closely related mammaliaforms since the Jurassic period. Here, we report the oldest record, to our knowledge, of hypsodont postcanines in the non-mammaliaform stem-mammal, *Menadon besairiei*, from the early Late Triassic. The post-canines are long and columnar, with open roots. They were not replaced in older individuals and remained functional after the total wear of the crown enamel. Dental histology suggests that, convergently to hypsodont mammals, wear was compensated by the prolonged growth of each postcanine, resulting in dentine hypsodont teeth most similar to extant xenarthran mammals. These findings highlight the constraints imposed by limited tooth replacement and tooth wear in the evolutionary trajectories of herbivorous mammals and stem-mammals.

[1] Programa de Pós-Graduação em Geociências, Instituto de Geociências, Universidade Federal do Rio Grande do Sul, Av. Bento Gonçalves, 9500, 91501-970, Bairro Agronomia, Porto Alegre, Rio Grande do Sul, Brazil. [2] Museu de Ciências Naturais, Fundação Zoobotânica do Rio Grande do Sul, Rua Dr Salvador França, 1427, 90690–000 Porto Alegre, Rio Grande do Sul, Brazil. [3] CONICET- Sección Paleontología de Vertebrados, Museo Argentino de Ciencias Naturales "Bernardino Rivadavia", Ave. Ángel Gallardo 470, C1405DJR CABA Buenos Aires, Argentina. [4] Departamento de Paleontologia e Estratigrafia, Instituto de Geociências, Universidade Federal do Rio Grande do Sul, Av. Bento Gonçalves, 9500, 91501-970, Bairro Agronomia, Porto Alegre, Rio Grande do Sul, Brazil. Correspondence and requests for materials should be addressed to T.P.M. (email: tomaz.melo@gmail.com)

Mammal reliance on their teeth for the processing of food is unparalleled among living vertebrates. While most other vertebrates have multiple generations of simple teeth, which are replaced continuously throughout the animal's life, mammalian dentitions are exceptionally complex and distinctive, with only two generations, the antemolar dentition being replaced only once, usually early in life (*i.e.* they are diphyodont as opposed to polyphyodont)[1,2]. Diphyodonty first evolved before the origin of crown-group Mammalia, in the closely related mammaliaforms[1,3,4]. It increased food processing efficiency via improved mastication, possibly leading to the evolution of endothermy, but it also exposed the now limited number of teeth to a higher degree of wear, particularly in abrasive herbivorous diets[5]. The necessity of improving dental durability in face of dietary and environmental changes has deeply influenced the evolutionary history of mammaliaforms, with multiple instances of convergence and parallelism[6], but other synapsids (stem-mammals) and some other extinct groups of non-mammalian vertebrates have independently acquired comparable adaptations against tooth wear[7].

Dental durability can be achieved in many ways, including modifications of enamel microstructure and thickness, delayed tooth eruption, addition of supernumerary teeth, increase in tooth height (hypsodonty), and often, a combination of methods[7,8]. While mammals employ all of these adaptations, supernumerary teeth are extremely rare, and hypsodonty is much more frequent[8]. Hypsodont postcanines, or cheek teeth, are associated to grazing, feeding in open habitats or at the ground level, and with diets rich in abrasive material, in which the wear of mineralized tissues is compensated by the ongoing tooth eruption[9]. The main source of the abrasive particles in the food consumed, intrinsic or extraneous, is subject to intense debate[10,11], although the importance of grit and volcanic ash has been receiving renewed recognition[12].

Hypsodonty is unknown outside Mammaliaformes, whereas large batteries of occluding teeth (*e.g.* in dinosaurs[13], captorhinid reptiles[14]) and enlarged molariforms with thick enamel (*e.g.* in diadectid stem-tetrapods[15], cynodont synapsids[16]) have evolved repeatedly[17–20]. Among stem-mammals, non-mammaliaform cynodonts were one the most diverse groups to successfully employ dental innovations for oral processing, in the form of complex heterodont dentitions, with occlusion of molariform postcanines evolving in some groups. Non-mammaliaform cynodonts appeared in the Late Permian[21] and became extinct by the Early Cretaceous[16], with peak diversity during the Triassic[22,23]. The group contains carnivores/insectivores (*e.g.* most probainognathians), omnivores and specialized herbivores[24] (*e.g.* traversodontids, tritylodontids[16]). During the Late Triassic, the first mammaliaforms[25–27] diverged from probainognathian non-mammaliaform cynodonts.

*Menadon besairiei* is nested within the clade Traversodontidae[28], a group non-mammaliaform cynodonts that developed relatively complex postcanine occlusion and derived tooth replacement independently and earlier than those of other cynodonts and mammals[24]. The labio-lingually expanded postcanines, subjected to extensive wear, are called gomphodont teeth. This dental pattern originated in the Early Triassic in more basal forms, the diademodontids and trirachodontids, which together with traversodontids constitute the clade Gomphodontia[24,28–30]. In the typical replacement pattern of traversodontids (continuous sequential replacement), the occlusion was not disrupted by vertical replacement of the gomphodont postcanines, because new teeth erupted sequentially only at the rear of the dental series, making the replacement primarily horizontal[24,31].

Based on previously undescribed material belonging to the traversodontid cynodont *Menadon besairiei* (early Late Triassic - Carnian - of Brazil[32] and Madagascar[33]), we report the presence of hypsodont postcanines and show that the patterns of dental growth and replacement were modified from the typical traversodontid pattern (low-crowned teeth with continuous sequential replacement). This unexpected convergence with mammals pulls the record of hypsodonty back in 70 million years. Although the particular conditions that led to the novel adaptation in this taxon cannot be easily determined, the unique cessation in postcanine replacement seems especially significant, in the context of the highly abrasive diet in the gomphodont lineage as a whole, during the hot and dry Triassic period.

## Results

**Description**. The new material of *Menadon* shows that the postcanines are columnar and open-rooted. Each tooth is curved, mesially in the upper and distally in the lower dentition, the same direction of their cant in implantation (Fig. 1; Supplementary Fig. 1). There is little differentiation between crown and root, and no cervix or transition can be found in the base of the extra-alveolar portion the postcanines. The intra-alveolar part of the teeth is about three times deeper than the extra-alveolar one, and the labio-lingual width is two to four times shorter than the coronal-apical height, depending on the tooth position (Supplementary Tables 1, 2). In well preserved teeth, the external surface is slightly rugose and continuous from the coronal wear facets to the edge of the pulp cavity (Fig. 1a–f). This overall morphology is indicative of hypsodonty.

To assess the implantation of postcanines, the intra-alveolar portion of a maxillary tooth row was exposed in the specimen UFRGS-PV-1165-T (Supplementary Note 1). It clearly shows the periodontal space, secondarily filled by minerals, around each tooth, implying a non-mineralized periodontal ligament. The unerupted or erupting distal postcanines, common to other traversodontids, are absent (Fig. 1i; Supplementary Note 2).

The cessation of sequential replacement is also observed in the lower dentition of all except the smallest, presumably youngest, specimens of *Menadon*. The last lower postcanine (but never the last upper postcanine) is remarkably reduced in size (Supplementary Table 3), marking the stop in eruption of additional distal postcanines.

Moreover, the layer of enamel[34] present in other traversodontids is absent in upper postcanines (PC) and absent or very reduced in lower postcanines (pc) of adult specimens (Fig. 1a–f), while enamel is clearly present in incisors and canines. *Menadon* possessed more extensive enamel in its postcanines at an earlier ontogenetic stage, as seen in the right?pc5 of a juvenile lower jaw (MCN PV-102016-T; approx. length 90 mm) (Supplementary Notes 1, 2; Supplementary Fig. 2c), but it was worn to the very thin band that can be preserved in adults, as on the labial ridges of the right pc2-4 in the larger lower jaw UFRGS-PV-0891-T (approx. length 170 mm) (Supplementary Note 1; Supplementary Figs. 1, 2a). This represents the minimal known amount of wear suffered by a mature individual.

**Histological analysis**. Histological analysis of the isolated postcanines (Supplementary Note 3) shows that the same mineralized tissues are distributed along the whole tooth in concentric layers (Fig. 2). Enamel was not found in the sections, instead, the peripheral layer is composed of cellular cementum, possessing abundant Sharpey's fibers, the imbedded portions of extrinsic collagen fibers (Fig. 2d, i), and cell lacunae left by cementocytes (Fig. 2a, d–h), over a thin layer of acellular cementum. The cementum layer extends to the apices of the cusps and ridges in the occlusal surface, and is thicker in the coronal portion. Internal to the cementum, the teeth are filled with orthodentine, which is

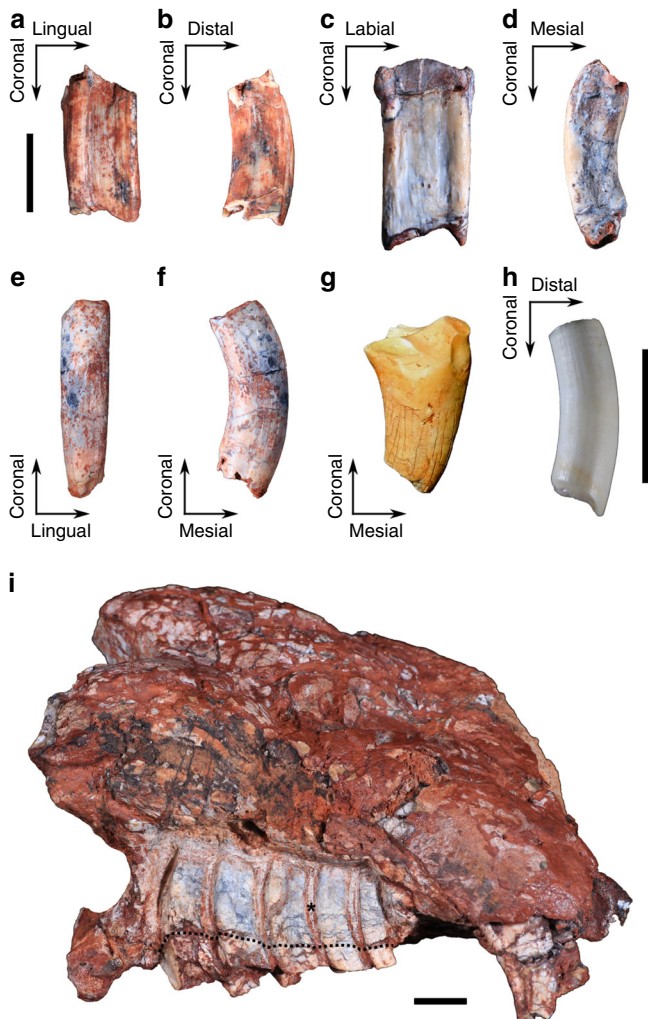

**Fig. 1** *Menadon besairiei* postcanine dentition. Upper postcanine UFRGS-PV-1333-T. **a** distal view; **b** labial view. Upper postcanine MCN-PV-10221 T, **c** distal view; **d** labial view. Lower postcanine MCN-PV-10343 T in, **e** mesial view; **f** labial view. **g** *Exaeretodon riograndensis* MCN-PV-3101 T. Heavily worn right lower postcanine in labial view. **h**, Extant xenarthran, three-toed sloth *Bradypus* sp. MCN-2771. Second upper molariform in lingual view. **i**, *Menadon besairiei* UFRGS-PV-1165-T. Right maxilla with the postcanine dentition exposed. Asterisk marks a location of periodontal space. Dotted line represents the original limits of the alveoli. Scale bars equal 10 mm, **a–g** are in the same scale

divided in distinct external and internal portions (Fig. 2b, f). The external dentine is richer in peritubular dentine and the tubules are noticeably more branched and bifurcated, while the internal dentine has more intertubular dentine and less branching (Fig. 2b). The pulp chamber is restricted to the root apical third of the postcanines, flaring apically (Fig. 2). Incremental lines in the dentine mirror the outline of the pulp chamber (Supplementary Fig. 3).

## Discussion

Although *Menadon* is placed, by dental and cranial characters, in the well supported clade of cynodont traversodontids[28,33,35], grouped with other specialized herbivores (*e.g. Exaeretodon* spp., *Scalenodontoides macrodontes*; Fig. 1g), the columnar morphology of its postcanines is, superficially, more similar to that of dentine hypsodont molariforms of xenarthran mammals[36,37], such as extant sloths and armadillos (Fig. 1h). Dentine

hypsodonty can be defined as a delay in root completion, coupled with prolonged formation of dentine[36,38], it often results in the fast wear of crown enamel and in lack of distinction between intra- and extra-alveolar portions[36,39], as occurs in *Menadon*. This interpretation is supported by our histological analysis, which confirms the presence of cementum, a tooth attachment tissue, around the occlusal areas near the tips of cusps and ridges of upper and lower postcanines, as it has long been associated to hypsodonty[37,39]. The small, but wide, pulp chamber and the incremental lines suggest progressive formation of dentine from coronal to root apical direction, instead of the centripetal infilling of the pulp chamber by secondary dentine of most synapsids[40,41], resembling the euhypsodont cheek-teeth of xenarthrans[42,43].

The Sharpey's fibers in the cementum (Fig. 2d, i) indicate a ligamentous attachment, the periodontal ligament, holding the tooth in the alveolus, constituting a gomphosis attachment[44,45]. Permanent gomphosis is regarded as the standard tooth attachment for extant mammals and archosaur reptiles[46], but has been shown to be a transient phase in the dental development of other tetrapods, before mineralization of the ligament and ankylosis of the tooth to the alveolar bone (*e.g.* refs [2,15,46,47]). Stem-mammals appear to have undergone this transition more than once[40,44,48], perhaps because the periodontal ligament allows for greater plasticity in the repositioning and migration of teeth[40,41]. The existence of a lasting gomphosis in *Menadon* is supported by the periodontal space between cementum and alveolar bone, which would contain the periodontal ligament in life. Also, crania with empty alveoli and isolated teeth of *Menadon* and other traversodontids are frequently collected[35,49–51], evidence that the periodontal ligament was decomposed and released the teeth from the sockets before burial[52]. Given the dynamic sequential replacement of the postcanines of traversodontids, in some taxa coupled with mesial drift[53] and extensive remodeling of the alveolar bone[54], as well as the continuous eruption and presence of cementum in *Menadon*, it is probable that all gomphodontian cynodonts maintained a lasting periodontal ligament throughout the life of each tooth[8,44].

In *M. besairiei*, differential wear between the softer internal intertubular and the harder external peritubular dentine[55] probably helped maintain the occlusal morphology after the enamel was worn away. Accordingly, almost the entirety of the cusps and ridges are composed of external dentine, plus cementum (Fig. 2). Some xenarthrans have a comparable arrangement of mineralized tissues in dentine hypsodont molariforms, though usually with more types of dentine, other than orthodentine, and cementum[37,56,57]. Other traversodontids and basal gomphodontians (*i.e.* diademodontids) can also lose most of the enamel on the anterior postcanines due to heavy wear, which changed the occlusal surface considerably[24,29,30,49] (Fig. 1g). In traversodontids, the postcanines erupted with much thinner enamel inside the occlusal basin than on the exterior, and on the lingual side the canines and incisors than on the labial side[24,49]. This indicates that differential wear was a strategy already being exploited by traversodontids, in which secondary occlusal surfaces were functional for prolonged periods and the incisors and canines were self-sharpening.

The tooth replacement in traversodontids is derived from their gomphodontian precursors. Incisors and canine teeth retained the plesiomorphic alternate pattern of replacement, whereas the postcanines developed a more complex sequential tooth succession. In early gomphodontians, such as diademodontids and trirachodontids, the gomphodont-type postcanines often coexisted with sectorial (cutting) and simpler postcanines. Eruption occurred sequentially from front to back of the tooth row, with the posterior sectorials, when present, being replaced by gomphodont or other sectorial teeth[24,29,31,51]. In the diademodontid

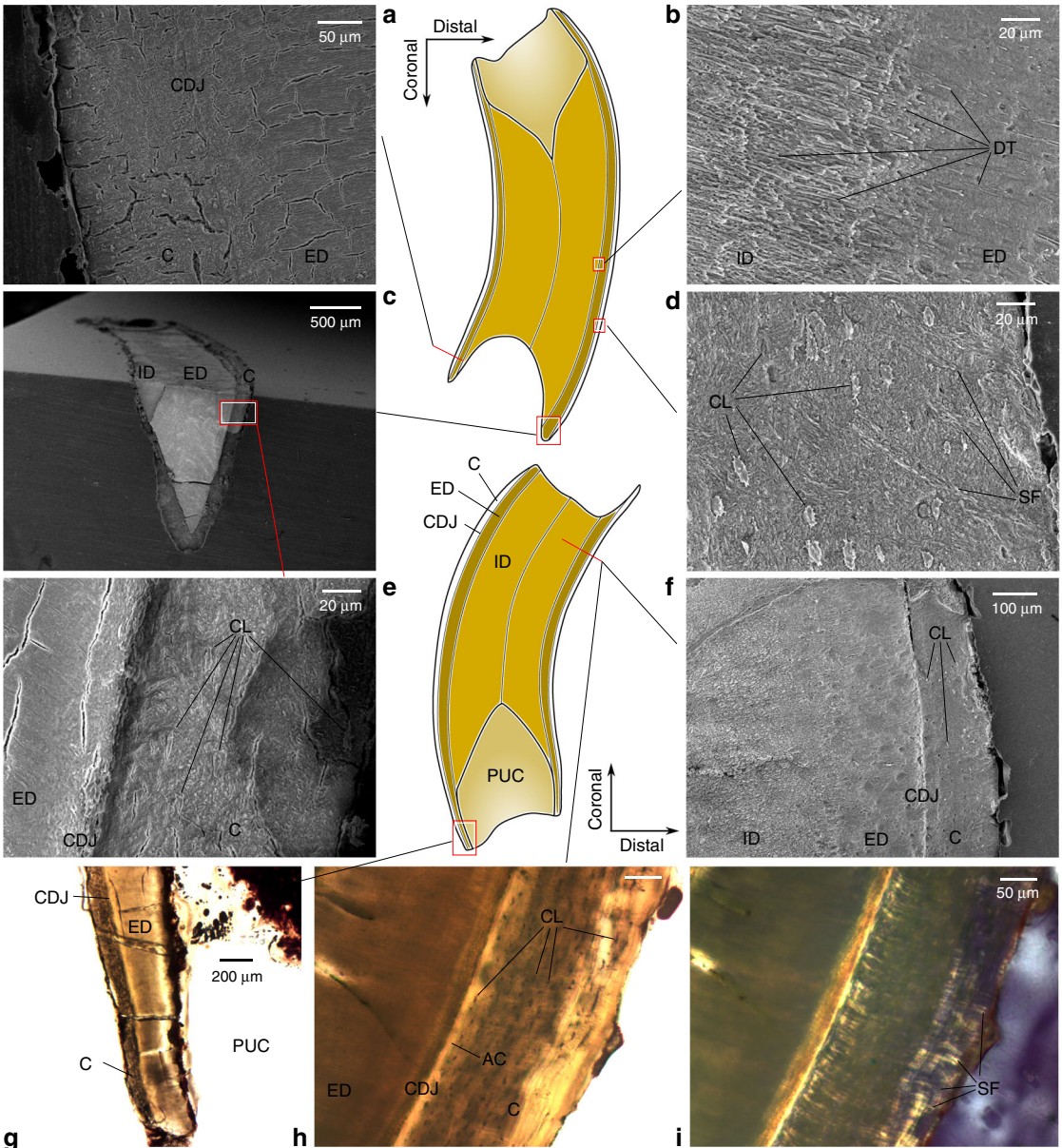

**Fig. 2** *Menadon besairiei* postcanine dental microstructure. **a**, **f**, **h**, **i**, cross sections; **b**, **d**, **g**, longitudinal sections; **c**, **e**, longitudinal (labio-lingual) section. AC acellular cementum, C cementum, CDJ cement–dentine junction, CL cementocytes lacunae (secondarily filled), DT dentine tubules, ED external dentine, ID internal dentine, PUC,pulp cavity, SF Sharpey fibers

*Diademodon*, at least three generations of postcanines were present at the same time, with distinct morphologies (conical, gomphodont, sectorial), continually replacing each other throughout the animal's life[29,58]. Trirachodontids, such as *Cricodon metabolus*, had sectorial and gomphodont postcanines as adults, but juvenile specimens show that an initial generation of cheek teeth was completely sectorial, followed by several waves of replacement[59,60]. Traversodontids generally lack any sectorial dentition as adults, although some species (*e.g. Andescynodon mendozensis*) had posterior sectorials as small juveniles, later replaced by gomphodont postcanines[61], and *Boreogomphodon jeffersoni* is the exception in retaining its sectorial postcanines through maturity[31]. In traversodontids with only gomphodont postcanines, the sequential replacement appears to represent a single generation of teeth, despite the potentially indefinite number of added teeth. In some species, notably *Exaeretodon* spp., the postcanines drifted mesially[49,53], pressed by the eruption

of distal postcanines, causing the oldest anterior postcanine to be lost, a process called Continuous Dental Replacement[8,53,62]. In such context, the stop in the replacement in *Menadon* is highly atypical from the gomphodontian and traversodontid patterns. (Fig. 3).

The dentition of *Menadon besairiei* reveals an unexpected set of dental characters, largely convergent to mammals, despite its precocity and distant phylogenetic relationship between the groups. It combines a very early example of hypsodont postcanines in the fossil record, with occlusion and heavy wear of tooth crowns, mammal-like gomphosis attachment and limited replacement. This novel record can help shed light on the evolutionary constraints on dental traits that appeared independently in multiple mammalian lineages, but very rarely among all the other vertebrates[7,36]. The oldest records of hypsodont postcanines, in mammaliaforms, date from the possibly euhypsodont Upper Jurassic (150 Ma) *Fruitafossor*[63], and later, Late Cretaceous

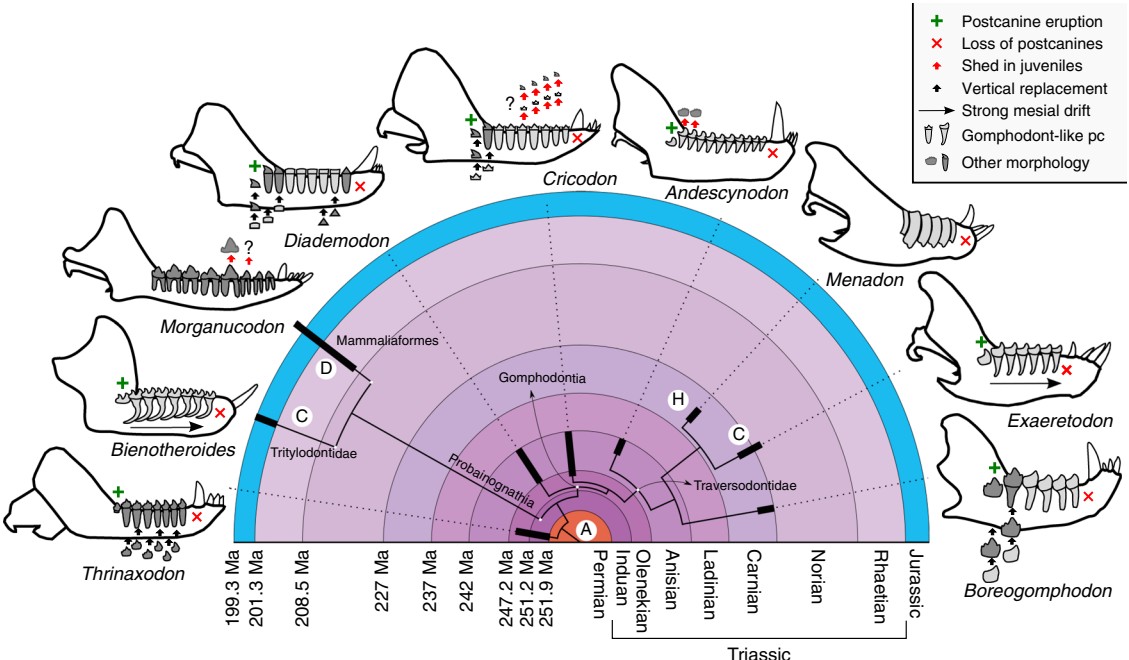

**Fig. 3** Postcanine replacement of Triassic and Early Jurassic cynodonts. Lower jaws of mature individuals of selected species. Green cross, addition of new tooth loci in adults; red "x", loss of older teeth in adults; Horizontal arrow, strong mesial drift[62]; black vertical arrows, continuous vertical replacement; red vertical arrows, deciduous teeth lost at a younger age and not replaced. *A* alternate replacement, *C* horizontal continuous sequential dental replacement of postcanines, *D* diphyodonty, *H* hypsodonty. Not to scale. Phylogeny based on Liu & Abdala[28] and Ruta et al.[22]. Chronostratigraphy from Cohen et al.[80]

(70 Ma) sudamericid gondwanatherians[64] with hypsodont cheek-teeth, and euhypsodont Eocene (55 Ma) xenarthrans[42] and taeniodonts[36]. Eleutherodont mammaliaforms from the Jurassic (165 Ma) and some Cretaceous (130 Ma) tritylodontid cynodonts have "dentine hypsodont" postcanines with high dentine walls formed by proximally fused roots[16,36,65], but we do not consider them equivalent because they were probably not exposed to wear.

Regardless of other differences, some lineages of herbivorous mammals became strikingly convergent with certain traversodontids in having a monophyodont dentition with continuous sequential replacement. Furthermore, as in traversodontids and other gomphodontians, related species of mammals often employ different solutions to the problem of dental durability, such as continuous sequential replacement in manatees of the genus *Trichechus*[8] and dentine hypsodonty in the dugong *Dugong dugon*, which also has teeth with enamel when they erupt and some degree of mesial drift[66]. In other cases, the same species employs more than one strategy, as in hypsodont elephantids that also have delayed sequential eruption, and the hypsodont mole-rat *Heliophobius argenteocinereus*, with continuous sequential replacement[8].

Convergently in early mammaliaforms and gomphodont cynodonts, the evolution of more precise occlusion led to drastic alterations in the replacement pattern (diphyodonty and continuous sequential replacement, respectively), which minimized misalignments between opposing teeth. In conjunction with the herbivorous diet consumed by traversodontids and other gomphodontians, this resulted in increased dental wear. We hypothesize that the determinate number of postcanines in *Menadon*, contrasting to all other traversodontids, constrained its evolutionary trajectory, favouring the appearance of hypsodonty, in detriment of strategies more commonly employed by polyphyodont species.

During the Middle and early Late Triassic, different lineages of gomphodont cynodonts experienced a reduction in postcanine crown complexity, by loss of extra tooth morphologies and

simplification of crests, cusps and cingula. At the same time, they developed a more complex interlocking mechanism between postcanines, larger occlusal surfaces by means of central occlusal basins, leaf-shaped or self-sharpening incisors, and optimized craniodental features for muscles related to food processing[24,28]. This period coincided with the diversification of the *Dicroidium* Flora in Gondwana[67], characterized by more rigid and resistant leaves than those of the preceding *Glossopteris* Flora, and with the strengthening of megamonsoonal climatic regime, marked by strong seasonality[68]. *Menadon* is part of a faunal assemblage (*Santacruzodon* Assemblage Zone; see Methods) older than the Carnian-Norian first occurring dinosaurs[35,69], after which earlier groups of herbivores, such as traversodontids and dicynodonts, were gradually replaced by dinosaurs, sphenodontians, and mammaliaforms[18,70] An arid palaeoenvironment is proposed for the unit, with high amounts of wind-blown dust[71,72], a potential abrasive that could accumulate on the vegetation consumed by *Menadon*[11,12,43].

In this context, *Menadon besairiei* represents a rare record of hypsodonty outside Mammaliaformes and highlights an unrecorded strategy to face the high dental wear imposed on specialized herbivores by the arid Triassic environments, and by the evolutionary constraints of the unusual mode of tooth replacement of gomphodont cynodonts. The specific conditions that led *Menadon* to develop such divergent morphology from all other traversodontids remain to be further investigated, as it is possible that underlying differences in morpho-functional aspects and dietary preferences between species might be essential to the explanation. Comparable adaptations would not be documented again for, at least, the next 70 million years, when diphyodont mammaliaforms diversified in dietary habits, during the Jurassic.

## Methods

**Dental terminology.** We refer as 'hypsodont' to teeth that have prolonged growth, resulting in increased height, particularly when the additional dental tissue is

submitted to wear. Since dentine hypsodont teeth often don't have clearly defined root and crown, we avoided these terms where it could cause confusion, instead using 'extra-alveolar' for the exposed coronal portion of the tooth and 'intra-alveolar' for the portion inside the alveolus. We followed the dental terminology proposed by Smith & Dodson[73].

**Material of *Menadon besairiei*.** All specimens of *Menadon besairiei* from Brazil were examined. The Brazilian material of *Menadon* derive from the same locality, the Schoenstatt outcrop, in Santa Cruz do Sul city, part of the *Santacruzodon* Assemblage Zone of the Santa Maria Supersequence[35,69]. Most fossils in the Schoenstatt outcrop are found disarticulated and isolated in the top five meters of massive and laminated red mudstone, with the predominance of skull and mandibles. Traversodontid cynodonts (i.e. *Santacruzodon hopsoni* and *Menadon besairiei*) are numerically dominant in the assemblage, with rare examples of carnivorous cynodonts[74] and archosauromorphs[75,76].

The Santa Maria Supersequence encompasses four distinct vertebrate associations (*Dinodontosaurus* AZ, *Santacruzodon* AZ, *Hyperodapedon* AZ and *Riograndia* AZ), ranging from?Ladinian-earliest Carnian to Norian[69]. The *Santacruzodon* AZ was dated as younger than 236,6 ± 1.5 Ma through detrital zircon U-Pb geochronology[32], which is concordant with the absolute ages for the underlying *Dinodontosaurus* AZ (236–233 Ma by correlation with the Chañares Formation from Argentina[77]) and the overlying *Hyperodapedon* AZ (younger than 233 Ma[78]; 231–225 Ma via correlation with the Ischigualasto Formation from Argentina[79]).

**Histological analysis.** One upper and one lower whole postcanines were selected to be microscopically analyzed (MCN-PV-10221 and 10343, respectively), as well as three fragments of the coronal portion of an additional upper postcanine (UFRGS-PV-1333-T). The fossils were chosen for their completeness and quality of preservation. They were sectioned in (partial) longitudinal and cross sections and, when possible, the counterparts resulted from the same cut were compared using light microscopy (LM) and scanning electron microscopy (SEM). The fragments were imaged only in SEM, they included a portion of the mesial wall (cross-section), the mesio-labial cusp (cross section) and the main disto-labial cusp (tangential section). For LM and SEM preparation, the teeth and tooth fragments were embedded in epoxy resin (Araldite GY 279). The material was then cut with a diamond saw and/or wet ground with 80 to 600 grit sandpaper. The samples for LM were glued on glass slides and thinned to approximately 100 μm. Subsequently, they were polished to 2500–3000 grit size to be viewed and imaged in a Zeiss Axio Scope.A1 light microscope using a Zeiss AxioCam ERc5s mounted camera. We used the 'photomerge' tool in Adobe Photoshop CC 2015 to reconstruct the entire sections.

The SEM samples were further polished to 3000 grit size and cleaned in an ultrasonic bath. Etching was performed with 1 N HCl for 120 s, followed by washing in deionized distilled water for five minutes and ultrasonic cleaning for 30 s. After drying, the samples where sputter-coated with gold and imaged (at magnifications between ×25 and ×3000) in the scanning electron microscope (JEOL JSM 6060) at a voltage of 10 kV.

**Reporting summary.** Further information on research design is available in the Nature Research Reporting Summary linked to this article.

## Data availability

The fossil material and histological sections described are deposited at the paleontological collections of the Instituto de Geociências, Universidade Federal do Rio Grande do Sul (UFRGS-PV) and Fundação Zoobotânica do Rio Grande do Sul (MCN-PV) in Porto Alegre, Brazil. The data supporting the findings of this study are available within the paper and its supplementary information.

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

## Acknowledgements

We thank C. Bertoni-Machado, D. Fortier, A. Liparini, T. Oliveira, T. Raugust C. Schultz, and L. Lopes for field and laboratory assistance. We thank J. Ferigolo, C. Schultz, L. Kerber, F. Abdala and V. Pitana (*in memorian*) for discussion. Funding was provided by Conselho Nacional de Desenvolvimento Científico e Tecnológico grants 141006/2015-3 (T.P.M.) and 304143/2012-0 (M.B.S.).

## Author contributions

T.P.M., A.M.R., and M.B.S. conceived the study. T.P.M. and A.M.R. prepared fossils and performed experiments. T.P.M., A.M.R., A.G.M., and M.B.S. contributed to analysis of results and discussion. T.P.M. prepared the manuscript, figures, tables and supplementary information. A.M.R., A.G.M., and M.B.S. commented and revised the text and figures.

## Additional information

**Competing interests:** The authors declare no competing interests.

