## [Peer Review File · Nature Communications]

Reviewers' comments:

Reviewer #1 (Remarks to the Author):

The study detailed in this manuscript shows very interesting findings and presents the earliest known report of prolonged growth of the dentition in a stem mammal (non-mammalian cynodonts) from the Triassic period. It pushes back the age of this innovation, which is widespread in mammals, to at least 70 million years. These findings can have interesting evolutionary and paleoecological implications worth to be mentioned and deeply discussed, and that is why they deserve to be published. However, there are many flaws in the presentation in this paper, which needs to be strongly improved, and also many imprecisions. More generally, the present results should be put in a broader evolutionary context, and discussed accordingly (with more references), to be accessible for a wider audience and be more convincing (especially the conclusion). As a result, the introduction, the results and the discussion should partly be reorganized and rewritten according to the suggestions detailed in the pdf file. Some interesting information and references from the "supplementary discussion" could also be included in the main text, especially regarding the brief review (but shortened) of tooth replacement in cynodonts. The manuscript should also be more accessible for non-specialist readers. The specific taxonomy and terminology used should be more importantly detailed or defined for more clarity.

Most of my comments and suggestions can be found in the attached file. I hope that they will significantly help the authors, and I will be glad to read a new version of this paper.

Helder GOMES RODRIGUES

Reviewer #2 (Remarks to the Author):

Comment to Melo et al.:

Earliest evidence of molariform hypsodonty in a Triassic stem.mammal.

Teeth are essential for animals to disintegrate food items for acquiring the needed energy for their life. Mammals evolved highly differentiated dentitions, whereas teeth of non- mammalian dentitions are often regarded as less sophisticated. Melo et al. describe a highly derived tooth structure in the traversodont cynodont Menadon from the Late Triassic of Brazil. These teeth show – surprisingly – a hypsodont structure. In mammalian dentitions hypsodont teeth were evolved several times independently and in different constructions.

The applied methods brought excellent result. In contrast to some other students, they restricted their studies not on the SEM only but combined light microscope and SEM work in a very efficient way. Their finds are original and well documented by the fossil material. The descriptions are clear focusing on the subject. The figures are very informative.

The paper of Melo et al. do not only state the hypsodonty for Menadon, but discuss its construction with those of mammals. Therefore, the article provides much more than new data but a very interesting contribution to the problem how different animals cope with the problem of destructive wear during mastication. The careful analysis shows that the teeth of Menadon have some enamel in their early stage, but continue as dentine teeth, thus the hypsodont teeth of Menadon are comparable to the mammalian type of dentine hypsodonty.

To aspects, seem to me of special interest. 1) The very early occurrence of hypsodonty already in the Late Triassic. 2) The impressive case of parallel evolution among non related vertebrates.

Therefore, I recommend the publication of the paper and its supplementary information warmly in the form they are now.

Response to Referees Letter

We would like to thank both reviewers for your consideration and for the helpful suggestions.

As requested, we have reorganized and rewritten the manuscript, focusing more on the evolutionary context and making it clearer and more accessible to readers. We looked to reduce the number of specialized terms throughout the text, defined the ones present at their first use and included clarification on the terminology in the Methods section. The Introduction was almost completely rewritten, with some information added from the Discussion section. The Results were streamlined by removing the interpretative parts and adding them to the Discussion. The Discussion section was restructured, with interpretation of the Results and comparisons to other taxa in the beginning (three paragraphs), an expanded discussion on the evolution of tooth replacement in gomphodonts (partly from the Supplementary Material), followed by discussions on early instances of hypsodonty in the fossil record and convergences between gomphodontians and extant mammals (slightly modified from the previous version). The final part concerns the palaeoecological context of the period, to which we added clearer statements about climate and a small section regarding age and the Carnian Pluvial Episode. We slightly modified the concluding paragraph to define more explicitly the conditions that probably led to hypsodonty in *Menadon*. The Methods section was expanded and subdivided to include information on dental terminology, background of the fossil material, and histological methodology. Finally, we added tooth orientations to the figures, and tentatively included taxonomic information and a legend box to fig. 3 for clarity.

We have followed most the specific suggestions and addressed the points raised by the reviewer #1 in the text and figures. Below are the answers to the questions and corrections we. If we overlooked any comment or failed to address it properly, please let us know so that we can correct it.

Line 1- Title. We maintained the word “molariform”, because dicynodonts, another group of stem-mammals, had dentine euhipodont tusks (caniniforms) since before the Triassic.

Lines 6-8 – Abstract: “Hypsodont cheek-teeth evolved independently in multiple mammalian lineages and in the closely related mammaliaforms, but have never been described in other vertebrates, or from before the Jurassic period.”

“Any evolutionary or environmental explanations? Limited set of teeth in hypsodont taxa (except mole-rats)? ...”

I hope to have made this point more clearly in the revised manuscript, but as it is a central aspect of the article, it always merits explanation...

In my opinion, the most likely explanation for the timing of the evolution of hypsodonty is more related to developmental constraints than to specific environmental changes. In mammaliaforms, dental occlusion, diphyodonty and oral processing of food are given, and I would expect hypsodonty to evolve when tooth wear becomes too high, as it has happened in so many mammal groups, be it because of drying climates, dietary changes or tectonics.

As most other tetrapods are less reliant on oral processing and can dispose of an indeterminate number of teeth (or do not use teeth at all for this purpose), wear is not such an important limiting factor. Gomphodont cynodonts are an exception in this regard, showing extreme wear in their postcanines, despite their (modified) polyphyodonty. Thus, it makes sense to me that, in this group, it might be advantageous to make the teeth more durable, instead of making more teeth that have to be moved around the mouth in order to be shed. Or at least, the balance between the two evolutionary strategies (durable X abundant teeth) might be more susceptible to influences by environmental factors or evolutionary novelties.

If diphyodonty first evolved during the latest Triassic-Early Jurassic in Mammaliaformes, earlier forms would more probably respond to these issues in other ways, rather than prolonging dental growth. Later (diphyodont) species would be more evolutionarily constrained and hypsodonty might be a more viable “option”. Of course, by the Jurassic, the ecological role of “grazer”, or medium to large-sized vertebrate herbivore, had been occupied by dinosaurs in most ecosystems, which probably limited the opportunities of hypsodonty to evolve in mammaliaforms, specially until the appearance of social insects and termitophagy/myrmecophagy (possibly around the time of *Fruitafossor*).

Lines 79-80 – Results: Histology. **“Meaning that you only sectioned adult teeth ? But what about the juvenile specimen described in supplementary data?”**

We only sectioned isolated teeth. The age of the individual can be hard to assess, since adult teeth vary in size depending on their position, but are otherwise very similar. The juvenile lower jaw is, so far, the only one ever found, so destructive analyses could not be performed.

Lines 81, 91, Fig 2 – Usage of “fibre” and “fiber”.

We maintained the use of “fibre”, as opposed to the American form “fiber”, because the rest of the text is written in British English. We actually were in doubt about the preferred variant of English, as we could not find this information in the Nature Communications website (and some Nature journals ask specifically for American or British English).

Lines 153-156 – Discussion. **“Studying the morpho-functional properties of the masticatory apparatus of Menadon compared to its sister taxa could be of high interest to understand the presence of hypsodonty (...)”**

Yes, I agree. Even though detailed morpho-functional analyses are beyond the scope of our study, a small amount of work has already been done in this regard (e.g. Ranivoharimanana, 2012), and I believe some more is currently underway by other research groups, hopefully they will bring better information on the sister taxa, which is also scarce. However, I can remark that the overall cranial/mandibular morphology in *Menadon* is quite similar to other traversodontids (e.g. fig. 6 of Ray, 2015). One difference seems to be the deeper skull (Melo *et al.*, 2015), likely to accommodate the elongated upper postcanines, but the dentary, for example, is not as deep as you could expect based on the different root shapes of other species, which are much shorter and narrower. Therefore, it would be interesting to understand the morpho-functional characteristics of other (more classical) gomphodonts in order to determine which parameters might be relevant for comparison, and what are their palaeobiological meaning.

Lines 153-156 – Discussion. **“(…) And you should slightly improve the quality of your diagrams (e.g. replacement of teeth in *Diademodon*, *Cricodon*…), which are not very clear.”**

I was not completely sure about how to improve the quality, so I updated the diagram and enlarged the teeth of some species, in case they were too small, and added a legend for the symbols. It also seems that the figure quality in the pdf is even more reduced than in the doc file, so I annexed the figures as separate files during submission.

We hope that we were able to improve the manuscript in relation to the last version and look forward to further feedback from the referees.

REVIEWERS' COMMENTS:

Reviewer #1 (Remarks to the Author):

The authors significantly improved their manuscript, especially the introduction, and they included most of my previous comments and suggestions. I think that even if this paper could be then accepted for publication, the abstract should be a bit more convincing for a wider audience. It also remains some imprecisions in text, and precisions are needed at some point for non-familiar readers, especially regarding the taxonomy. Most of my other comments and suggestions are listed in the pdf, but they are mainly related to minor issues.

I also wish to answer one comment from the authors regarding the evolution of hypsodonty in the present taxon:

Authors: "In my opinion, the most likely explanation for the timing of the evolution of hypsodonty is more related to developmental constraints than to specific environmental changes. In mammaliaforms, dental occlusion, diphyodonty and oral processing of food are given, and I would expect hypsodonty to evolve when tooth wear becomes too high, as it has happened in so many mammal groups, be it because of drying climates, dietary changes or tectonics..."

My answer: I rather think that increasing the longevity of the dentition in synapsids is tightly related to environmental factors in relation to increasing dental wear. Herbivores generally have more complex teeth to improve plant comminution, but these teeth are difficult to be continuously vertically replaced in relation to developmental constraints or issues (e.g. high energetic cost?). As a result, there are two possibilities to increase the efficiency of the dentition: hypsodonty or continuous horizontal replacement. This latter alternative is generally frequent in polyphyodont synapsids such as traversodontids and tritylodontids, whereas hypsodonty is widespread in mammals. However, some studies, including the present one, showed that hypsodonty can occur in synapsids previously polyphyodont, and continuous horizontal replacement in diphyodont mammals, and both strategies can also occur in the same species (e.g. the silvery mole-rat). The evolutionary and developmental explanations regarding the unexpected presence of these innovations in this group remains to be discovered. The best example is the sirenians, in which the dugong have a degenerate hypsodont dentition, while the manatees have a continuous horizontal replacement. It is thus difficult to explain why close taxa, with a very similar developmental background, present two different dental strategies in relation to grazing. The same question arises concerning Menadon compared to other traversodontids, meaning that there is no simple adaptive explanation, and that should be further investigated.

I would be glad to further discuss these aspects if needed.

Helder GOMES RODRIGUES

Response to Referees Letter

Again, we would like to thank the referee for the helpful suggestions and discussion.

We have followed the suggestions and addressed the points raised by the reviewer in the text and figures. Below are all the reviewer's comments (in bold) and the answers to the questions and corrections that demanded explanation or were not promptly altered in the manuscript. If we overlooked any comment or failed to address it properly, please let us know so that we can correct it.

“The authors significantly improved their manuscript, especially the introduction, and they included most of my previous comments and suggestions. I think that even if this paper could be then accepted for publication, the abstract should be a bit more convincing for a wider audience. It also remains some imprecisions in text, and precisions are needed at some point for non-familiar readers, especially regarding the taxonomy. Most of my other comments and suggestions are listed in the pdf, but they are mainly related to minor issues.

I also wish to answer one comment from the authors regarding the evolution of hypsodonty in the present taxon:

Authors: “In my opinion, the most likely explanation for the timing of the evolution of hypsodonty is more related to developmental constraints than to specific environmental changes. In mammaliaforms, dental occlusion, diphyodonty and oral processing of food are given, and I would expect hypsodonty to evolve when tooth wear becomes too high, as it has happened in so many mammal groups, be it because of drying climates, dietary changes or tectonics...”

My answer: I rather think that increasing the longevity of the dentition in synapsids is tightly related to environmental factors in relation to increasing dental wear. Herbivores generally have more complex teeth to improve plant comminution, but these teeth are difficult to be continuously vertically replaced in relation to developmental constraints or issues (e.g. high energetic cost?). As a result, there are two possibilities to increase the efficiency of the dentition: hypsodonty or continuous horizontal replacement. This latter alternative is generally frequent in polyphyodont synapsids such as traversodontids and

tritylodontids, whereas hypsodonty is widespread in mammals. However, some studies, including the present one, showed that hypsodonty can occur in synapsids previously polyphyodont, and continuous horizontal replacement in diphyodont mammals, and both strategies can also occur in the same species (e.g. the silvery mole-rat). The evolutionary and developmental explanations regarding the unexpected presence of these innovations in this group remains to be discovered. The best example is the sirenians, in which the dugong have a degenerate hypsodont dentition, while the manatees have a continuous horizontal replacement. It is thus difficult to explain why close taxa, with a very similar developmental background, present two different dental strategies in relation to grazing. The same question arises concerning *Menadon* compared to other traversodontids, meaning that there is no simple adaptive explanation, and that should be further investigated.

I would be glad to further discuss these aspects if needed.”

Yes, I absolutely agree. I was addressing the specific points of the rarity/absence of hypsodonty before the rise of mammaliaforms, and what are the prerequisites or important factors for the evolution of this trait. Simple adaptive explanations are often not satisfactory, even for modern groups with known diets and physiologies, and with better temporal and phylogenetic resolution. I was assuming that, although tightly linked to the evolution of hypsodonty, the environment changed considerably over geological time, only leading to hypsodonty in few clades, and was speculating on what these groups have in common.

Line 8 – Not necessary

Lines 13-14 – You should rather emphasize the convergence with crown mammals (i.e. xenarthrans) based on dental histology.

Line 20 – Include only

Lines 22-23 – Not necessary.

Line 24 – I would have said "increased food processing efficiency via improved mastication".

Line 30 – Which adaptations?

Line 31 – to->of

**Line 44 – Do you mean large batteries of occluding teeth and thick enamel?
Precise, please.**

Line – 51-52 You should add precisions on the mode of replacement.

Explained in end of the same paragraph.

**Line 56 – Why the occlusion would be disrupted by vertical replacement of teeth?
Is not the case in mammals? (see other comment p.7)**

Yes, but in the case of mammals, it happens once per locus (at most), usually early in life, with many lineages losing the deciduous dentition altogether, or shedding it *in utero*. Also, the fast and determinate growth of mammaliaforms means that a permanent dentition will not become obsolete as the skull continues to grow (Luo, *et al.*, 2004; O'Meara & Asher, 2016). This is not true for the majority of vertebrates, which are polyphyodont, with continuous alternate replacement that does not allow for the establishment of occlusal relations between the teeth, affecting the performance in occlusion (Crompton & Jenkins 1968). For comparison, an iguana will replace its teeth about five times **per year**, while replacement in the basal cynodont *Thrinaxodon* is estimated to have occurred four to seven times during life (Abdala *et al.* 2013).

In lineages that developed a form of dental occlusion that depends on correspondence between opposing teeth, it is frequent the suppression of replacements. There are several instances of squamates and sphenodontians with acrodont teeth, displaying occlusion and cessation of replacement (Nydam *et al.*, 2000; Zaher & Rieppel, 1999), as the agamid lizards, with distally erupting acrodont teeth that are never replaced (Cooper *et al.* 1970).

To my understanding, this is the most accepted hypothesis for the evolution of diphyodonty in mammaliaforms. Please see my reply to the comments of p.7.

Lines 61-62 – Precise please, for non-familiar readers.

Detailed in the previous paragraph.

Lines 87-88 – Ok, you are right. But for more accuracy, the dental height vs length should be measured.

These measurements are presented in the Supplementary Tables.

Line 99 – And thus youngest...

Line 102 – -typical

Line 102 – around the crown->present in

Line 129 – Why superficially?

Because they are shaped similarly, and have the same general disposition of tissues (dentine), but xenarthrans have osteodentine, vasodentine, and other specializations unique to the clade. This is mentioned in a later paragraph.

Line 139 – I do not think rodent incisor is a suitable example (because of the presence of enamel), contrary to the cheek-teeth of xenarthrans.

Lines 152-153 – I think that there are more appropriate and recent references regarding mesial drift, and bone remodelling. E.g.

- **Wise GE, King GJ (2008) Mechanisms of tooth eruption and orthodontic tooth movement. *Journal of Dental Research* 87: 414–434.**
- **Gomes Rodrigues H, Solé F, Charles C, Tafforeau P, Vianey-Liaud M, Viriot L. 2012. Evolutionary and biological implications of dental mesial drift in rodents: the case of the Ctenodactylidae (Rodentia, Mammalia). *PLoS ONE* 7:e50197**

Altered accordingly.

Line 153 – Did you observe such a remodelling in your specimen and in other traversodontids?

The surface of the alveolar bone, or the bone surrounding the postcanines, is usually more porous and has a different aspect in comparison with other parts of the maxilla or dentary. It is visible in most well-preserved specimens of many species, especially in empty alveoli and inter-alveolar septa, but not particularly in *Menadon*. I was referring mainly to the species with strong mesial drift, such as *Exaeretodon* spp. I changed the text slightly to better convey this idea.

Lines 184-186 – I think it is difficult to talk about monophyodonty in the case of horizontal replacement, given that this term was generally associated to a vertical replacement. Note that in each case, it corresponds to an extension of the dental lamina to form new teeth. To avoid problems, you should not mention monophyodonty here, as in your figure 3 (or only for Menadon).

I agree. The definitions can be ambiguous in this regard, and by the most common ones (*e.g.* Bertin *et al.* 2018), some traversodontids could be both mono **and** polyphyodont.

Line 187 – You should cite more appropriate references such as

- **Abdala et al., 2002 on Exaeretodon dental replacement**

- or
- **Gomes Rodrigues & Sumbera 2015 on dental drift (and CDR) in traversodontids. (sorry for the self citations, but they are relevant here).**

Included.

Line 192 – You should rather say "convergent"

Lines 199-201 – It could be difficult for non-specialist of cynodonts (or mammals) to visualize these taxa and their evolution without a phylogeny. Just add a few precisions (age...) for each taxa, please.

Line 208 – Note that enamel is also present when teeth erupt and mesial drift also occur in dugongs.

Line 215 – This is most probable that rapid and intense wear precludes malocclusion and misalignment (as in mammals having CDR), but not the horizontal replacement. So, I do not think that modifications of dental replacement in traversodontids is highly convergent to mammaliaforms in this aspect (in relation to "precise" occlusion).

I am not sure what you mean in the first sentence. Are you saying intense wear prevents malocclusion in mammals in general, including those that have CDR? And that horizontal replacement does not prevent malocclusion? Maybe I should clarify that I was referring to the initial evolution of mammaliaforms and their defining characters, which most likely happened during the Triassic. Horizontal replacement is not really true replacement, but rather sequential eruption, without subsequent waves of replacement.

What I wanted to say was that, in both (early) mammaliaforms and gomphodonts, occlusion probably comes first, and then changes in replacement (in both cases, sequential replacement and fewer vertical replacements, as I discussed in a previous answer). Fewer replacements lead to more wear. Herbivory also leads to more wear. Too much wear impairs occlusion and mastication (although some wear might be necessary to shape matching surfaces; see Crompton & Jenkins, 1968 and Crompton, 1972). In other words, intense wear reduces the durability of dentitions, unless there are adaptations in place that help to cope with it (Janis & Fortelius, 1988).

This sequence is seen in mammal-lineage cynodonts (Probainognathia) such as *Pachygenelus* (polyphyodonty, alternate replacement, incipient occlusion and wear facets), and then in the mammaliaforms, *Sinoconodon* (partial diphyodonty, sequential replacement and incipient occlusion), *Morganucodon* (determinate growth, diphyodonty, sequential replacement, more precise occlusion and matching wear facets) and so on, with many other specialisations appearing, before crown mammals (Crompton & Jenkins 1968; Crompton, 1972; Luo et al., 2004). Note that all the species mentioned above are considered insectivorous.

As for the degree of convergence between groups (early gomphodonts and early mammaliaforms), I agree that the convergence is not exact in the details, but the underlying evolutionary strategies seem to be very similar. This idea has been pointed out in the literature, for example, Crompton and Jenkins (1968; footnote in page 445) state:

“It is interesting to note that in both the gomphodont cynodonts and mammals which develop **complex occlusion**, alternate tooth replacement was independently lost. This, in part, accounts for the similarities and differences between dental succession in gomphodonts and mammals.”

I apologise for the long answer, specially if I misunderstood your comment.

Line 217 – The reverse hypothesis could also be suggested: the development of hypsodonty favour a reduction of the dentition and limit the replacement as in some extant mammals.

Yes, the reverse hypothesis is not unfeasible. However, I believe it is more unlikely.

Of the many groups of mammals that developed hypsodonty, only a fraction became monophyodont afterward. On the other hand, many brachyodont species reduced replacements or became monophyodont, for various reasons. Therefore, at least for mammals, that already have a plesiomorphic determinate number of teeth, it does not seem that one is a prerequisite for the other, even there is some influence. In the mammalian case, the determinate dentition undoubtedly evolved first, in Triassic mammaliaforms.

In traversodontids, most of the history of this transition is still unknown.

Considering its rarity, the presence of hypsodonty outside Mammaliaformes seems to demand more explaining than the comparingly mundane limitation of replacement/eruption. In analogy with extant mammals, the reverse hypothesis would be like a species with CDR (*e.g.* a manatee) evolving some degree of hypsodonty and later reversing to a determinate number of teeth, while my original hypothesis would be more similar to a mammal attaining a determinate number of teeth (as occurred) and then evolving some degree of hypsodonty.

Fig. 3 – Did you define the degree of mesial drift based on Gomes Rodrigues et al., 2010 and Gomes Rodrigues & Sumbera, 2015? Add precisions, please.

Added a Gomes Rodrigues & Sumbera (2015) as reference.

Fig. 3 – Where are the stippled horizontal arrows in the figure?

It was removed from a previous version. I have corrected the legend.

Fig. 3 – Mention that the replacement is horizontal.

Line 234 – I do not really understand the relation between the CPE and Menadon, since it probably occurred well after the setting of hypsodonty in this taxa. You should only focus on the environmental changes as well the arid paleoenvironmental conditions you mentioned for Menadon.

Line 236 – Do you know if a fossorial hypothesis has been proposed for this taxa based on cranial remains? Hypsodont could also represent an adaptation related to this specialization.

Not for *Menadon* and not on cranial remains, but for other gomphodont cynodonts, it was proposed based on postcranial material and small cynodonts have been found inside burrows, including the gomphodontian *Trirachodon* (Groenewald *et al.*, 2001). *Menadon* and gomphodontians, in general, tended to be larger than other cynodonts, and the known burrows from the Triassic are usually too small. There is no record of burrows in the *Santacruzodon* Assemblage Zone.

Lines 237-240 – You can also suggest further analysis (FEA, dental microtextures...) to explain the different dental "strategies" between traversodontids, in relation to the morpho-functional aspects of their masticatory apparatus and the putative partitioning (or differences) of food resources. But see also my general comment concerning occurrence of hypsodonty vs continuous horizontal replacement in synapsids (in response to your previous comment).

Line 248 – Especially in the case of dentine hypsodonty.

We hope to have further improved the manuscript based on your suggestions.